# Redox Signaling from Mitochondria: Signal Propagation and Its Targets

**DOI:** 10.3390/biom10010093

**Published:** 2020-01-06

**Authors:** Petr Ježek, Blanka Holendová, Lydie Plecitá-Hlavatá

**Affiliations:** Department of Mitochondrial Physiology, No. 75, Institute of Physiology of the Czech Academy of Sciences, Videnska 1083, 14220 Prague, Czech Republic; blanka.holendova@fgu.cas.cz (B.H.); lydie.plecita@fgu.cas.cz (L.P.-H.)

**Keywords:** Redox signaling from mitochondria, mitochondrial superoxide formation, H_2_O_2_ diffusion, peroxiredoxins, HIF, redox-regulation of kinases

## Abstract

Progress in mass spectroscopy of posttranslational oxidative modifications has enabled researchers to experimentally verify the concept of redox signaling. We focus here on redox signaling originating from mitochondria under physiological situations, discussing mechanisms of transient redox burst in mitochondria, as well as the possible ways to transfer such redox signals to specific extramitochondrial targets. A role of peroxiredoxins is described which enables redox relay to other targets. Examples of mitochondrial redox signaling are discussed: initiation of hypoxia-inducible factor (HIF) responses; retrograde redox signaling to PGC1α during exercise in skeletal muscle; redox signaling in innate immune cells; redox stimulation of insulin secretion, and other physiological situations.

## 1. Preface

Recent findings documented the existence of redox signaling affecting numerous biological processes in the cell [1,2,3,4,5,6]. This statement is not trivial, since one may ask how it is possible that such a rather non-specific transient burst of reactive oxygen species (ROS) can specifically affect certain targets. The key to understanding lies in uncovering mechanisms of ROS production in various physiological situations and mechanisms of how the local “signal” given by the local appearance of particular ROS species is spread and transferred to the target. Of course, identification of a target under particular conditions is also not trivial. Moreover, one can accept a concept of passive targets, i.e., all proteins affected along the way of redox signal spreading. These passively targeted proteins can be distinguished from “final targets”, i.e., one or several proteins that are affected so that particular cell function is significantly altered.

Focusing on mitochondrial ROS sources initiating redox signaling [7,8,9], one may distinguish (*i*) intramitochondrial redox signaling, when all targets are located within the structure of mitochondrial tubular network; and (*ii*) redox signaling from mitochondria to targets located in the cytosol, nucleus (retrograde redox signaling) or even targets existing within the extracellular matrix. Examples of the most studied physiological redox signaling from mitochondria are (1) initiation of hypoxia-inducible factor (HIF) and consequent transcriptome reprogramming; (2) retrograde redox signaling to PGC1α during exercise in skeletal muscle; (3) redox signaling in innate immune cells; (4) redox stimulation of insulin secretion due to metabolism of secretagogues such as branched-chain keto acids and fatty acids; and other situations, which we attempt to describe in this review. However, details are still unknown concerning the redox signal transfer from the sources of superoxide (as a prevalent initiator in these cases) in mitochondria to the identified targets. Moreover, under conditions when mitochondrial ROS sources are elevated even further, pathological oxidative stress is induced, which can be multiplied by the amplification of cytosolic ROS sources. Thus, redox signaling from mitochondria is essential, but when exaggerated it substantiates further oxidative stress, and thus amplification is induced, leading to pathology (Table 1).

That is why in this review we discuss the two major determinants of the redox signaling: (*i*) how the transient redox burst is possible at all; and (*ii*) what are mechanisms and ways of transfer of such redox signals to proper targets. Current knowledge, as well as hypotheses, are reviewed. One could also describe certain effects of ROS under conditions of mild to severe oxidative stress as pathological redox signaling. However, due to the vast area of this topic, we will mention this only scarcely.

## 2. Elevations of Mitochondrial Superoxide Formation

### 2.1. Mechanisms of Mitochondrial Superoxide Generation

Excellent reviews can be recommended to recall mechanisms of superoxide generation within the respiratory chain complexes, but also by several mitochondrial dehydrogenases, as well as electron-transfer flavoprotein: coenzyme Q oxidoreductase (ETFQOR) [10,11,12,13]. Here we shall discuss under which conditions elevations of superoxide formation may occur, which subsequently could represent sources for redox signals. Note that alternatively a sudden decrease of antioxidant activity might also initiate a redox signal at constant superoxide formation. However, this mechanism is less probable in mitochondria. Nevertheless, as an example, one may recall a sudden inhibition of uncoupling provided otherwise by the mitochondrial uncoupling proteins (UCP) [14,15]. Such an inhibition would result in providing a slight surplus of superoxide, since before the inhibition of UCPs, its formation is suppressed by uncoupling of respiratory chain proton pumping from the proton backflow through the c-ring of the mitochondrial ATP-synthase.

Inevitably, redox homeostases are always related to metabolism. The first typical mechanism of elevation of mitochondrial superoxide formation is concerned with Complex I of the respiratory chain. It is executed upon a sudden increase of substrate of the Krebs cycle (turnover), or other metabolites usable for respiration and oxidative phosphorylation (OXPHOS). A surplus of respiration is required to elevate the NADH/NAD^+^ ratio (that may be termed as a substrate pressure), which results in an increase of superoxide formation in the flavin site I_F_ of Complex I of the respiratory chain [16,17,18]. With various substrates and in different cell types, also site I_Q_ of Complex I produces superoxide, as located in the vicinity of the ubiquinone binding site. With relative slowdown of ubiquinone (coenzyme Q) cycling, concomitantly retarded electron transfer through Complex I allows for higher superoxide formation [18].

The second mechanism lies in the retardation of cytochrome *c* cycling between the Complex III and Complex IV (cytochrome *c* oxidase), which effectively slows down the electron transfer so that superoxide formation can be elevated at the outer (proximal to the intracristal lumen) ubiquinone site within the Complex III, termed III_Qo_ [10,40,41]. One may predict that this mechanism is inevitable upon apoptotic initiation when cytochrome *c* migrates out of the intracristal space lumen. However, upon a sudden impact of hypoxia, this mechanism is initiated in an as yet unknown way. This is the key mechanism of redox signaling transferred to the prolyl hydroxylase domain (PHD) enzymes (alternatively termed EGLN), which leads to one of the ways of HIF1α stabilization and concomitant HIF-mediated transcriptome reprogramming.

The third mechanism stems again from the superoxide formation at the ubiquinone site of Complex I (I_Q_); however, it occurs upon the reverse electron transport (RET), mediated by ubiquinone within the inner mitochondrial membrane (IMM) [42]. Instead of transferring electrons from Complex I or Complex II (succinate dehydrogenase, SDH) to Complex III, RET is defined as electron flow back from Complex II to the Complex I. Thus, RET can be initiated namely in situations of succinate accumulation, such as during reperfusion after hypoxia [35] and metabolic transitions in brown adipose tissue (BAT) [36,43,44].

Mechanisms of how certain dehydrogenases in the mitochondrial matrix can form superoxide are not well understood. Their capability to contribute to mitochondrial superoxide formation was judged from experiments with isolated mitochondria [10], as well as in the case of α-glycerolphosphate dehydrogenase, located probably at the outer (intracristal lumen) surface of IMM (this cristae portion of IMM is also termed intracristal membrane, while lumen is termed intracristal space, ICS).

The fourth established mechanism of enhanced superoxide formation in mitochondria is executed upon β-oxidation of fatty acids [45] or β-like oxidation of branched-chain ketoacids, metabolites of branched-chain amino acids. In both cases, ETFQOR at its elevated turnover forms a surplus of superoxide [10].

### 2.2. The Interplay between ROS, Mitochondrial Anion Channels, and Mitochondrial Permeability Transition

Under pathological conditions, intra- and extra-cellular ROS also affect mitochondrial proteins through redox-dependent post-translational modifications. This may be further amplified by mitochondrial ROS generating systems. As a result, excessive ROS are subsequently released from mitochondria to the cytosol [46]. Specifically, mitochondrial ion channels may influence mitochondrial redox homeostasis as they influence the electric component of protonmotive force Δp, established by proton pumping of the respiratory chain from the matrix to ICS. Such a component is termed mitochondrial membrane potential for simplicity (Δ*Ψ*_m_). The whole protonmotive force is consumed physiologically by H^+^ backflow from the ICS back to the matrix via the c-ring of membrane F_O_-sector of ATP-synthase. When active, mitochondrial uncoupling proteins partly or substantially consume Δp (its both components, Δ*Ψ*_m_ and ΔpH) and attenuate Δ*Ψ*_m_-dependent superoxide formation [14,15]. However, non-zero membrane H^+^ permeability termed as H^+^ leak permanently acts against the established Δp. Its magnitude is reflected by the so-called non-phosphorylating respiration of mitochondria.

In general, ion channels typically receive cell signals from three distinct sources through the following: (*i*) cytosolic Ca^2+^ elevation; (*ii*) phosphorylation by Ca^2+^- and/or redox-dependent kinases, either cytosolic ones or those recruited to the mitochondrial matrix or ICS; (*iii*) other oxidative post-translational modifications dependent directly on cytosolic ROS; and, hypothetically (*iv*) other post-translational modifications such as acetylation, acylation, etc. Such signals alter the function of mitochondrial ion channels and they are also influenced mutually, e.g., upon variations of activities of Ca^2+^ and K^+^ influx mechanisms residing within IMM. Note also that accumulation of Ca^2+^ into the mitochondrial matrix directly affects efficiency of several dehydrogenases of the Krebs cycle, as well as the electron transfer through the respiratory chain, and hence ATP synthesis. Of course, all these factors influence the superoxide generation.

Besides the plain H_2_O_2_ diffusion (see below), superoxide anion may be exported from the mitochondrial matrix through specific mitochondrial anion channels such as the inner membrane anion channel IMAC [47,48,49,50]. This channel is still poorly characterized. Also, superoxide oscillations have been reported based on the IMAC opening [51,52].

A self-standing field concerns with the mitochondrial permeability transition pore (mPTP), largely dependent on calcium homeostasis. Recently, mPTP was suggested to act as a non-physiological specific conformation of the ATP synthase and its oligomers [53]. However, this topic is beyond the scope of this review. We have to only remain that long-standing disputes concern with the identity of this phenomenon. Both IMAC and mPTP can trigger ROS-induced ROS release in neighboring mitochondria (mitochondrial fragments) activating cell-death signaling [54].

### 2.3. The Interplay between ROS and Mitochondrial Ca^2+^ Uniporter and Mitochondrial Ca^2+^ Antiporters

Ca^2+^ influx into mitochondria is ensured by the mitochondrial Ca^2+^ uniporter (MCU), which has long been considered to be the only structure allowing mitochondrial Ca^2+^ uptake. However, other Ca^2+^ channels, such as rapid mode of uptake (RAM) [55,56], Leucine zipper-EF hand containing transmembrane protein 1 (LETM1) [57], CoQ10 [58], and mitochondrial ryanodine receptor type 1 (mRyR1) [59,60,61] were also discovered. These channels exhibit different properties relative to the MCU. When Ca^2+^ is transferred to the mitochondrial matrix down along its electrochemical gradient and without transport of other ions, there is a net transfer of two positive charges into the matrix resulting in a substantial drop of Δ*Ψ*_m_. [62,63] Ca^2+^ accumulation is prevented by action of mitochondrial 2Na^+^/Ca^2+^ antiporter. These proteins together with Na^+^/H^+^ antiporter ensure the overall Ca^2+^ homeostasis in the mitochondrial matrix [64].

Moreover, the Ca^2+^-stimulated respiration will not only compensate for the loss of Δ*Ψ*_m_ by the accelerated respiratory chain proton pumping, but also produces a net gain of ATP. Thus, the physiological range of Ca^2+^ uptake into the mitochondrial matrix stimulates ATP synthesis without a loss of Δ*Ψ*_m_. However, if sustained cytosolic Ca^2+^ concentration elevation occurs, the Ca^2+^ concentration triggers an excessive mitochondrial Ca^2+^ uptake called mitochondrial Ca^2+^ overload. This is followed by Δ*Ψ*_m_ depolarization, decreased ATP production, and, most notably, by an acceleration of superoxide generation, leading to activation of cell death pathways under various pathological conditions [46]. Exact molecular mechanisms underlying the Ca^2+^ overload and ROS generation are still unknown.

Recently, the activity of human MCU, the pore component of the uniporter complex, was detected to be regulated by redox post-translational modification S-glutathionylation at conserved Cys-97. The conjugation of glutathione causes a conformational change within the N-terminal domain and seems to promote MCU channel activity. Since the N-terminal domain faces the matrix, this region could serve as a sensor of matrix superoxide or H_2_O_2_ and subsequently cause the reprogramming of the Ca^2+^-dependent mitochondrial metabolic signals [65].

### 2.4. ROS and Mitochondrial K^+^ Channels

Mitochondrial potassium channels comprise a diverse group of ion channels with different properties, such as voltage-gated K^+^ channels, Ca^2+^ activated K^+^ channels, ATP-sensitive K^+^ channels and two-pore K^+^ channels [66,67]. These channels mediate K^+^ influx from the cytosol to the mitochondrial matrix following the electrochemical gradient of K^+^ [68]. A constant influx of K^+^ ions would cause not the only collapse of Δ*Ψ*_m_, but also mitochondrial swelling due to the osmotic entrance of water through aquaporins. To prevent such events, the K^+^ positive flux is counterbalanced by the K^+^ efflux exchanged for H^+^ on K^+^/H^+^ antiporter [69,70,71]. Such maintenance of K^+^ homeostasis and mitochondrial volume takes place at the expense of protonmotive force, and thus influences ATP synthesis, mitochondrial respiration, and ROS production [67,72]. Current evidence shows a possible modulation of the mitoK_ATP_ and mitoK_Ca_ in cardiomyocytes by the action of protein kinases PKA and PKC. The phosphorylation of the mitochondrial K^+^ channels during ischemic preconditioning seems to be a key mechanism in cardioprotection [73].

### 2.5. Mitochondrial ROS and Voltage-Dependent Anion-Selective Channels

Voltage-dependent anion-selective channels (VDACs) are channels (porins) of the outer mitochondrial membrane (OMM), mediating transport of ions and metabolites through the OMM. They are also able to interact with several cytoplasmic proteins, e.g., hexokinase [74], tubulin [75,76], dynein [77], actin [78], and others. All these factors enable the key function of VDAC in cell physiological processes with participation of mitochondria and typically during the initiation of apoptosis [79]. VDAC was confirmed to exist in three different isoforms VDAC1 to VDAC3, having high sequence similarity and structural homology, but differing in functions within the cell. VDAC1 was proposed to have a pro-apoptotic function [80]. In contrast, VDAC2 exerts an anti-apoptotic function [81] and VDAC3 was suggested to play a role in the control of ROS [82]. Several posttranslational modifications were detected within the VDACs’structure, such as phosphorylation [83], acetylation [84], and tyrosine nitration [85]. Recently, studies of posttranslational modifications of VDACs’ concerning cysteines and methionines uncovered the over-oxidation of certain Cys residues to sulfonic acid and Met to methionine sulfoxide. Interestingly, the authors showed that the number of oxidized residues in VDAC1 and 2 is relatively small compared to those in VDAC3. The differences might be assigned to different regulations of activity of these pore proteins [79].

## 3. Redox Signal Spreading Out of Mitochondria

Recently, excellent studies revealed that redox signals can be traced as instantly oxidatively modified cysteine residues which are spread via different sets of proteins in different tissues [86,87,88,89,90]. In other words, one may consider most of them passive targets and conclude that these are different in varying cells and tissue types. However, it remains to be established how the redox signal is transferred, e.g., within the >500 nm distance from the surface of OMM of the mitochondrial network tubules to the plasma membrane target protein (Figure 1). The simplest mechanism would be represented by a simple H_2_O_2_ diffusion over such a distance and subsequent direct oxidation of the target protein (in this example located at the plasma membrane) (Figure 2). Superoxide would unlikely diffuse to such a distance, and hence, if superoxide-mediated redox signaling exists, it could only affect proteins in the vicinity of OMM.

Alternatively, a mediated oxidation of target protein through the action of thiol-based proteins capable of redox relay to the target, such as peroxiredoxins (regenerated via thioredoxins and glutaredoxins), would provide a common signal transfer (Figure 2). However, it can be speculated whether a chain of peroxiredoxin oligomers itself could transfer the redox signal instead of the H_2_O_2_ diffusion (Figure 2, rightmost scheme). This would be theoretically possible if the internal S-S bridges (two intermonomer S-S bridges within peroxiredoxin homodimer, see below) could be reduced not by thioredoxins or glutaredoxins, but by the other neighbor peroxiredoxin. Hence, it is yet to be established whether a redox relay exists via an array of peroxiredoxin oligomers.

In the case of redox signaling from mitochondria, we must inspect various methods by which the initially elevated superoxide reaches the cytosolic compartment before discussing how the redox signal is transferred within the cell cytosol; or subsequently from the cytosol to the nucleus, to the endoplasmic reticulum (ER) or the extracellular matrix (ECM). The other aspects include questions as to whether a net signaling by superoxide is possible at all or whether superoxide must always be converted to H_2_O_2_ by superoxide dismutases, MnSOD in the matrix and CuZnSOD localized within the intermembrane space, i.e., between the tubular surface of the mitochondrial network given by OMM and so-called inner boundary membrane (IBM, i.e., a portion of IMM forming a parallel inner cylinder with OMM). CuZnSOD is probably also located in ICS.

### 3.1. Hypothetical Redox Signaling by Superoxide Diffusion?

Neglecting still hypothetical superoxide anion penetration and oscillations of superoxide local concentrations with the participation of IMAC (see above), one can consider that the superoxide release to the mitochondrial matrix can provide a redox signal only when converted to the membrane-permeant H_2_O_2_. Nevertheless, at least a portion of superoxide formed at the site III_Qo_ is released to the ICS lumen. Subsequent superoxide diffusion via crista outlets, if it exists, would be complicated. Thus, already superoxide diffusion is unlikely to the peripheral intermembrane space (forming a tubular sandwich with the OMM and IBM). This is because OPA1 heterotrimers, MICOS complexes, and other proteins guarding the crista outlets (crista junctions with OMM). Another barrier is represented by the OMM itself. Here, superoxide was suggested to penetrate via VDAC and similar pores or channels. In conclusion, it must still be established whether the simplest but physiological redox signaling mediated by superoxide diffusion from mitochondria exists. Note, however, that upon initiation of apoptosis the crista outlets are open, allow even cytochrome *c* migration out of ICS membranes and hence losses of cytochrome *c*, so then also superoxide can leak out (see [13] and references therein).

Two O_2_^•−^ molecules can spontaneously dismutate at a very slow rate, which is dependent on pH since protonation is a part of such dismutation [91]. Thus half-life of dismutation at 1 μM O_2_^•−^ is 2 s at pH 7 [92].

Note, that a portion of superoxide exists in the form of hydroperoxyl radical HO_2_^•^ (pKa 4.5). Hence, the ratio of O_2_^•−^ to HO_2_^•^ accounts for 1000:1 at pH 7.8 [12]. At lower concentrations such as existing in cells, but in a non-cellular environment a half-life of O_2_^•−^ might be in the order of hours. Moreover, thiol adduction to superoxide is as non-efficient as dismutation [93]. Consequently, the diffusion distance of O_2_^•−^ in the presence of SOD is 0.5 μm [94].

### 3.2. H_2_O_2_ Routes up to OMM

Thus, it is plausible that redox signaling from all mitochondrial sources is rather provided by the diffusion of H_2_O_2_ either through IBM plus OMM when matrix-released superoxide is converted by MnSOD, or via crista outlets plus OMM, when ICS-released superoxide is converted by CuZnSOD. One may also consider mediation through mitochondrial peroxiredoxins (PRDX3 and PRDX5, see below) and glutathione peroxidase GPX1. Besides further reducing H_2_O_2_ [95], they may speculatively help to spread mitochondrial redox signals at least towards the outer surface of OMM as an alternative to the diffusion of H_2_O_2_. Nevertheless, details of locations of PRDX3 and PRDX5 within ICS, peripheral intermembrane space between OMM, and IMB are not known.

## 4. Redox Signal Spreading within the Cytosol

### 4.1. Diffusion of H_2_O_2_

Theoretically, a direct diffusion of H_2_O_2_ to the targets may be the simplest way of redox signaling from mitochondria. At the vicinity of the target protein, H_2_O_2_ may either directly oxidize certain amino acid residues of the target protein or affect the target protein indirectly via peroxiredoxins. Interestingly, recent reports based on cytosolic H_2_O_2_-selective probes receiving matrix-originating artificially generated H_2_O_2_ showed that these cytosolic fluorescent probes were affected only when some cytosolic thioredoxins were ablated [96]. If confirmed, this would suggest the participation of peroxiredoxins in the spreading of the redox signal beyond the OMM surface. However, since redox relay between a chain of peroxiredoxins has not yet been confirmed to exist, this problem has yet to be studied.

H_2_O_2_ exerts a low reactivity and rather a long half-life. These properties lead to rather large diffusion-distances of ~0.5 μm in an aqueous environment [94]. The half-life of H_2_O_2_ in cells was estimated to be 10 to 1000 μs, as controlled by enzyme-catalyzed pathways [97]. Thus, e.g., peroxiredoxin exhibits a second-order rate constant of 10^8^ M^−1^s^−1^ for reaction with H_2_O_2_ [98].

### 4.2. Reactions of Thiol-Containing Proteins

Protein thiols can react with H_2_O_2_ to form sulfenic acid (R-SOH) by two-electron oxidation. Resulting sulfenylation is reversible and typically changes the conformation and/or activity of proteins. If a nearby thiol exists, sulfenic acid further reacts with such a protein thiol or thiol of reduced glutathione (GSH), to form an inter-/intra-molecular disulfide bridge (S-S) or protein-S-GSH disulfide, respectively [99,100]. Thus, sulfenic form (-SOH) can create a disulfide (-S-S-) bond with other proteins containing oxidized thiols into the sulfenic form.

The one-electron oxidation of a thiol also exists, giving thiyl radicals, which are subsequently transformed into a plethora of downstream oxidation products, such as S-nitrosothiols (SNO) and persulfides (S-SH) [101]. Similarly, methionine can be converted to methionine sulfoxide or methionine sulfone by oxidation [102]. Note also that oxidative stress is established when lysine, arginine, threonine, and proline residues undergo carbonylation, which is irreversible.

### 4.3. Peroxiredoxin Family

Peroxiredoxins (PRDXs) convey their oxidation by H_2_O_2_ to the terminal target protein, typically as phosphatases or transcription factors [103]. Thus, peroxiredoxins are capable of a literal “redox kiss” to the target protein. Other thiol-reactive systems are given by couples of thioredoxin and thioredoxin reductase (TRX and TRXR) or GSH/glutathione reductase (GR). These systems cooperate with PRDX and GPX enzymes to maintain them available in the active and reduced form. In this way, the reversibility of redox modifications of PRDX and GPX enzymes is established [100]. Thus, PRDXs as sensitive intracellular peroxidases coordinate cell (mitochondrial) redox signaling, since they act as local redox sensors also governing the binding affinity of partner proteins.

In human cells, six isoforms of PRDXs exist. PRDX1,2 and 6 are localized in the cytosol and nucleus, PRDX4 in the endoplasmic reticulum, while PRDX3 is exclusively mitochondrial, whereas PRDX5 is also located in mitochondria, cytosol, and peroxisomes [104]. Reaction mechanism depends on two cysteines in family members PRDX1 to PRDX 4 (2-cys-PRDX); a similar but atypical mechanism is performed by PRDX5, whereas dependence only on a single cysteine exists for PRDX6 (1-cys-PRDX) [105]. The active site of PRDXs Cys is highly reactive to H_2_O_2_. The 2-cys-PRDXs can be oxidized not only to sulfinic acid, but are irreversibly inactivated upon further oxidation to sulfonic acid [103,106,107]. In contrast, PRDX5 and PRDX6 are more resistant to sulfinylation. Typically, the resulting sulfenic acid residue forms a disulfide bond with the active site Cys of a partner within a homodimer [108]. Moreover, PRDX1,2 homodimers associate to decamers, in a doughnut-like structure of five homodimers, which is however destabilized when a disulfide bond is formed [103]. Finally, the cycle is completed by reduction of the two disulfide bonds of the homodimer catalyzed either by TRX or GRX enzymes.

Upon reaction with H_2_O_2_ PRDX5 forms intramolecular (intramonomer) disulfide bond, instead of intermonomer (intermolecular) disulfide bonds of PRDX1 to 4 [107]. The reaction is rather slow, having a rate constant of 10^5^ M^−1^∙s^−1^. Human PRDX5 also reacts with peroxynitrite (ONOO^−^; rate constant of 10^7^ M^−1^∙s^−1^). One can, therefore, predict that in a cell compartment such as mitochondrial matrix or cytosol other PRDXs outcompete PRDX5, which may then more specifically react with ONOO^−^ or even lipid peroxides.

PRDX6 is a 1-cys-PRDX forming homodimer, but being unable to form disulfide bonds. Instead, sulfenic acid is formed when this cysteine is oxidized. The sulfenic moiety of PRDX6 is then reduced with GSH, but not with thioredoxins [109]. In this way, PRDX6 reduces oxidized phospholipids. Interestingly, PRDX6 exerts also Ca^2+^-independent phospholipase A2 activity.

### 4.4. Floodgate Model

Properties of PRDXs enable them to be in the first front of redox signaling, as well as antioxidant protection. Indeed, PRDXs exert a high affinity for H_2_O_2_ determined by the Thr-Cys-Arg in their strictly conserved active site. This triad stabilizes the transition state by polarizing the O-O peroxyl bond [110]. As a result, the transfer of electrons from the catalytic cysteine toward peroxyl takes place, providing reaction rates of 10^5^–10^8^ M^−1^∙s^−1^ [111]. Indeed, this is several orders of magnitude higher than a general reaction of a protein thiol with H_2_O_2_ [112].

The local increases in H_2_O_2_ as an initiating event of redox signaling can be considered to act indirectly on target proteins through PRDXs according to the so-called floodgate model [2,113]. According to this model, scavenging PRDX enzymes are inactivated by H_2_O_2_ (oxidized to sulfinic and sulfonic moieties, see above). Therefore, only the remaining PRDXs existing in a sulfenic form may subsequently oxidize the passive or terminal target. Thus, for example, local increases in H_2_O_2_ may inhibit local peroxiredoxins and only a remote peroxiredoxin is capable to relay the redox signal to the target protein (Figure 2). Consequently, at higher H_2_O_2_ concentration (local) levels, if instead of sulfenic acid (R-SOH), higher oxidized states, sulfinic (RSO_2_H) moieties are formed, they may be regenerated i.e., reversed to sulfenic moieties by sulfiredoxin (SRX) enzyme. However, the reduction by SRX is much slower than that by TRX [114]. Moreover, when sulfonic (RSO_3_H) residues are formed, this is irreversible. As a result, their formation can be regarded as oxidative stress and not physiological redox signaling [115].

Another complex role of PRDXs lies in their ability to form high molecular weight complexes, i.e., oligomers of decamers described above. They are formed mostly upon higher peroxidation states (containing sulfenic or sulfonic moieties) and thus they loose peroxidase activity and the only chaperone-like activity is restored [116].

### 4.5. Signaling via Redox Relay

Redox relay from PRDX1to4 towards certain targets has already been described in some pertinent cases [117,118,119]. For example, the redox sensing function is ensured by the thiol-disulfide exchange reaction of PRDX2 with the transcription factor STAT3 to repress transcriptional activation in the nucleus [120]. This redox relay possesses an advantage of lower H_2_O_2_ concentrations being required for such redox signaling when compared to a simple H_2_O_2_ diffusion to the target, which is the signal transducer and activator of transcription 3 (STAT3).

Cytosolic PRDX1 also acts within a redox relay [121]. This is evidenced upon its ablation (or PRDX2 ablation), where lower amounts of oxidated cytosolic protein thiols are globally found [122,123]. For example, oxidized PRDX1 blocks the activity of apoptosis signal-regulating kinase 1 (ASK1), a Ser/Thr kinase, inducing apoptosis [121,124]. Interestingly, TRX forms also complexes with ASK1 within the same site but increased H_2_O_2_ dissociates TRX from ASK1 and thus activates it [125]. Similarly PRDX4 and GPX7 act within ER [126,127]. Note also that phosphorylation of PRDXs, which inhibits their function, is involved in the complex crosstalk between kinases, phosphatases, and PRDXs, which is beyond the scope of this review [3].

### 4.6. Mitochondrial Peroxiredoxins

After being imported to the mitochondrial matrix, a cleavage of 61-amino acids of the mitochondrial targeting N-terminal sequence in translated PRDX3 results in 21.5-kDa monomeric protein [128]. PRDX5 contains both, a 52-aminoacids long mitochondrial targeting N-terminal sequence and C-terminal SQL sequence addressing to peroxisomes [129,130]. Cleavage in the mitochondrial matrix results in a 17-kDa monomer.

### 4.7. Glutathione Peroxidases

GPX family consists of five enzymes with seleno-cysteine active sites (GPX1 to 4, GPX6) utilizing GSH as a cofactor [131,132]; and other three enzymes with a redox sensor role (GPX5, GPX7, GPX8) having only cysteine residues in their active sites and modest peroxidase activity [133]. The cytosolic and mitochondrial GPX1 and plasma membrane and cytosolic GPX4 are abundant in all tissues and cell types.

### 4.8. Amplification of Cytosolic ROS Production (Signaling) by Stimulation of NADPH Oxidases by Mitochondrial ROS

The potential cross-talk between mitochondrial ROS and NADPH oxidase (NOX) has long been known [134]. Mitochondrial ROS can be amplified by cytoplasmic NOX and vice versa, leading to feed-forward process that augments the pro-oxidative status required for pathological signaling; for example, during the development of pulmonary hypertension. Redox signaling from mitochondria also targets NADPH oxidases such as NOX1 and NOX2 to activate them. In endothelial cells, when such redox signaling is exaggerated the resulting oxidative stress contributes to hypertension. Similarly, in the development of pulmonary hypertension mitochondrial ROS activate NOXs and to collectively induce a pro-oxidative redox state, which is further favored by the impairment of antioxidant capacity (SOD, catalase and glutathione peroxidase) of pulmonary hypertensive cells [135].

The revealed mechanism for endothelial cells includes the proto-oncogene tyrosine-protein kinase c-Src redox dependent pathway [26]. Interestingly, mitochondrial K_ATP_ channel opening was related to such elevated mitochondrial superoxide formation, which subsequently potentiated elevation of cytosolic ROS via activation of NADPH oxidases [136]. Also, RET-dependent redox signaling activated NOX2 in human aortic endothelial cells [72]. Specifically, AngII, a peptide hormone being a key effector of the renin angiotensin system was shown to induce mitochondrial dysfunction by inhibiting the activity of PGC-1α [137]. Its downregulation reduces catalase expression through FoxO1 transcription factor, thus weakening antioxidant system [138]. Moreover, dysfunctional mitochondria produce mitochondrial ROS via inhibited electron transport chain and ROS together with accumulated Zinc then activates NF-κB-dependent upregulation of NOX1, causing telomere attrition and replicative senescence especially in vascular wall [139]. Mitochondria derived superoxide was also shown to activate NOX2, subsequently causing oxidative stress leading to hypertension [72]. Activation of the mitoK_ATP_ and subsequent matrix alkalization induced superoxide production by reverse electron transport of respiratory chain as malate supplementation reduced blood pressure [26]. A critical role of mitochondrial ROS for activation of NOX1 in various hypertension models was nicely summarized by Daiber et al. [140,141].

## 5. Mitochondrial Redox Signaling at Hypoxia

In the next sections, we describe the best known examples of mitochondrial signaling. The representative list and possible pathological consequences of the exaggeration and/or impairment of such a signaling are summarized in Table 1.

### 5.1. Hypoxia-Inducible Factor

Oxygen sensing in cells is provided by multiple mechanisms, among which central mechanism lies in action of 2OG-dependent prolyl hydroxylases (PHDs), alternatively termed Egl nine homolog 1 proteins (EGLN) [19,142,143]. PHDs provide a ferrous iron- (Fe^II^-) dependent plus O_2_-dependent plus 2-oxoglutarate-dependent hydroxylation of HIF-1α (or HIF-2α which is constantly degraded. Therefore, by decreased O_2_, by lowering the other PHD co-factors and by the increased ROS, PHDs are also inhibited and hence the HIF system is activated [20] (Figure 3). The system is more complex, since also factor inhibiting HIF (FIH) hydroxylates HIFα, but at different sites. Both, PHD and FIH are affected by ROS, hence participate in redox signaling.

Iron is acting as a co-factor in PHD reaction. Originally, it was considered that ROS oxidize ferrous iron (Fe^II^) to ferric iron (Fe^III^) [144]. Recently, also reactive cysteines were recognized in PHD2. Their oxidation inhibits PHD, hence initiates HIF-response upon oxidation [145]. A suggested mechanism may include the redox-induced formation of inactive PHD homodimers due to formation of disulfide bridges [146,147]. As a result, PRDX involvement in HIF activation may also be predicted (Figure 3). Hence, it can be also hypothesized that any redox signaling either from mitochondrial sources or redox signaling of non-mitochondrial origin should inhibit PHDs and stabilize HIFα.

PHD-mediated proline hydroxylation is required to set HIF-α susceptible for ubiquitination by the ubiquitin ligase, termed Von Hippel-Lindau tumor suppressor protein (pVHL) [148,149]. In this way, pVHL ensures the proteasomal degradation of HIF-α. The active function of PHDs and pVHL thus belong to the central mechanism allowing a constant degradation of HIF-1α (but also HIF-2α or HIF-3α isoforms) [21]. Upon inactivation of PHDs, HIF-α stabilization is induced. The stabilized HIF-α binds to HIF-1β/ARNT (or to HIF-2β, HIF-3β) and provides the reprogramming of transcriptome typical for the complex HIF system, i.e., hypoxic inductions of over 400 genes [150,151]. The hallmark of HIF-mediated transition is the promotion of Warburg phenotype by upregulation of pyruvate dehydrogenase kinase [152,153,154,155].

### 5.2. Role of Mitochondrial Redox Signaling in Hypoxic Adaptation

Ongoing debates throughout the years have been concerned with the question of how important the dependence of HIF system on mitochondria [19,156] is. It has been shown that PHDs sense oxygen independently of mitochondria. Nevertheless, mitochondrial metabolism and redox signaling represent a key player in HIF signaling. Indeed, PHDs are inhibited by the lack of oxygen but also by the lack of metabolites such as fumarate, succinate, malate isocitrate and lactate [27,157,158]. An essential requirement of mitochondrial redox signaling was suggested by the restoration of Δ*Ψ*_m_, which also reestablished superoxide formation; and was observed to restore hypoxic HIF1α stabilization in cells when respiration and hence Krebs cycle turnover was largely abolished [159]. The cells used for these experiments had deleted mitochondrial DNA polymerase, and hence lacked essential components of the respiratory chain and possessed an incomplete (vestigial) ATP-synthase. When the inhibitory factor IF1 of the ATP-synthase was also present, the reversed mode of this enzyme was blocked and Δ*Ψ*_m_ was nearly abolished, hence also a majority of superoxide formation. However, Δ*Ψ*_m_ was re-established upon IF1 ablation. This enabled the ATPase activity when ATP-synthase serves as a proton pump and establishes Δ*Ψ*_m_ [159].

In any case, inhibition of PHDs is essential for HIF responses and might be also ensured simply either by competitive inhibition of PHD cofactors by the accumulated Krebs cycle metabolites [158,160,161] or metabolites such as L-2-hydroxyglutarate [162]; or by the lack of PHD co-substrate 2OG. Thus, variations in 2OG levels given by mitochondrial metabolism play also a key role in controlling PHDs and hence the HIF system [163]. Note also that one of the plethora adaptive responses to hypoxia-mediated by HIF system is an elevation of the mitochondria-specific autophagy, termed mitophagy [164], as well as the influence of expression and structural organization of Complex I and Complex III, which in turn regulate superoxide formation [19]. Some of these aspects lead to the self-acceleration of mitochondrial superoxide formation. However, the extensive mitophagy would cancel the mitochondria-controlled HIF responses.

### 5.3. Mechanism of Complex III Initiated Mitochondrial Redox Signaling in Hypoxic Adaptation

Upon the impact of sudden hypoxia, a hypoxic ROS burst was observed [165], but delayed by several hours [166,167]. Other reports described a hypoxic ROS burst occurring immediately within a minute in endothelial, HeLa, and HK2 cells [168]. Originally, the Complex III site III_Qo_ has been identified as a source of such a hypoxic ROS burst [165,169,170,171,172,173,174]. A similar hypoxic ROS burst was reported to occur even during normoxic HIF activation [175]. Thus redox signaling provided by H_2_O_2_ emanates from mitochondria and oxidizes iron Fe^II^ in PHDs. This mechanism is among many others which lead to HIF-α stabilization.

The influence of the HIF system by the mitochondrial redox signaling was discovered based on experiments, when components of Complex III were ablated, which prevented HIF-1α stabilization [165], unlike in anoxia [170]. For example, the ablation of Rieske iron-sulfur protein stabilized HIF [22]. Moreover, suppressors of site III_Qo_ electron leak (S3QELs) prevented the HIF response [176].

However, the detailed mechanism is unknown to explain how a certain sensor of hypoxia affects the mitochondrial respiratory chain so that it takes several hours for the maximum HIF-α stabilization to occur [167]. The observed matrix superoxide burst precisely coincides with the maximum HIF-1α stabilization [166]. Hypothetically, a certain ICS redox buffer is overcome after the several-hr time period, so that the ROS burst at site III_Qo_ begins at that moment. One may speculate that certain components may act as a redox buffer in the outer IMS between OMM and IBM, such as the mitochondrial intermembrane space import and assembly protein 40 (MIA40) and the augmenter of liver regeneration (ARL) [177]. Oxidized ARL has been reported to be regenerated simply by oxygen, which is not possible at hypoxia when ARL was suggested to donate electrons to the cytochrome c, which effectively retards the electron transfer from the Complex III to the Complex IV.

Alternatively, oxygen dissolved within the intracristal membrane has to be exhausted first before slow down of the Complex IV (cytochrome *c* oxidase) reaction leads to a slow down of the cytochrome *c* cycling and inevitable elevation of superoxide formation at site III_Qo_. Note, that the partition coefficient of O_2_ in the lipid bilayer is ~4, hence despite its lack within the aqueous compartments oxygen can still participate in reactions within the membranes until it is exhausted also from the lipid bilayer.

Experiments using peroxiredoxin-5 overexpression in IMS exhibited attenuation of hypoxic ROS signaling [174]. This outcome supports the concept of exhaustion of a redox buffer within IMS during hypoxic initiation of HIF-α stabilization. Similarly, redox-sensitive GFPs addressed to IMS/ICS locations responded to ongoing hypoxic redox signaling [172].

The instant retardation of electron flow beyond the Rieske iron-sulfur protein due to hypoxia has not yet been explained. In contrast, a HIF-mediated switch (delayed) between the “normoxic isoform” of cytochrome c oxidase subunit-4 (COX4.1) and the COX4.2 “hypoxic isoform” has been described [153]. However, this presents us with a “chicken-and-egg” situation, since the observed redox burst should precede and initiate the HIF-mediated signaling.

### 5.4. Mechanism of Complex I Initiated Mitochondrial Redox Signaling in Hypoxic Adaptation

A knockdown of Complex I subunit NDUFA13 (GRIM-19) leads to increased superoxide formation which subsequently causes HIF1α stabilization plus accelerated autophagy [178,179]. Since the HIF activation depends exclusively on the loss of the SDHB subunit [180], which contains the iron-sulfur cluster, RET and hence Complex I_Q_ site is a probable source of superoxide in this situation. Since major ablations of respiratory chain Complex III subunits, such as of Rieske iron-sulfur protein impair and restructure the whole respiratory chain and its supercomplexes, one may consider that also Complex I-generated superoxide participates in HIF activation under these conditions [181]. Also specific inhibitor of Complex I prevented HIF1α stabilization [182].

Even termination of hypoxic signaling may be considered to exist as feedback from the resulting HIF-mediated transcription reprogramming. This can exist since the Complex I subunit NDUFA4L2 is a HIF-target gene [183]. Its induction not only decreased respiration but paradoxically diminished also superoxide formation [184]. In general, one can consider that upon higher ubiquinone-H_2_(ubiquinol)/ubiquinone ratio (i.e., CoQH_2_/CoQ), which is directly proportional to the electron transfer efficiency, low superoxide is formed and vice versa [185].

## 6. Mitochondrial Redox Signaling in Skeletal Muscle

### 6.1. Exercise Evoked Signaling in Skeletal Muscle

During acute exercise three major types of intracellular signaling are induced: increases in cytosolic Ca^2+^, increase in ATP turnover and mitochondrial redox signaling, i.e., emanation of ROS from mitochondrion [30,186]. All three types are integrated so that they increase transcriptional activity of the master regulator of mitochondrial biogenesis, peroxisome proliferator-activated receptor-gamma coactivator-1α (PGC1α) (Figure 4). While Ca^2+^ acts via calcium-calmodulin kinase (CaMK) [187], the AMP/ATP ratio stimulates the AMP-activated protein kinase (AMPK) pathway [188], and elevations of mitochondrial ROS represent redox signals targeting p38 mitogen-activated protein kinase (MAPK) pathway. The latter is followed by phosphorylation of transcription factors MEF2 and ATF2, which in turn regulate the expression of PGC1α [189]. H_2_O_2_ was also shown to promote directly the activity of PGC1α promoter. Exactly this was the E-box location within this promoter since mutation of the E-box largely prevented the effects of H_2_O_2_ in inducing expression of PGC1α [190,191]. Exercise also recruits p53 into the mitochondrial matrix to bind a non-coding D-loop of mitochondrial DNA (mtDNA) and TFAM, a transcription and structural factor essential for mtDNA [192].

### 6.2. Exercise Evoked Mitochondrial Signaling Targets PGC1α in Skeletal Muscle

PGC1α serves as a coactivator of peroxisome proliferator-activated receptors PPARγ and PPARα, estrogen-related receptors (ERR), thyroid hormone receptor TRβ, and nuclear respiratory factor 1 (NRF1) plus nuclear factor erythroid 2-related factors (NRF2). Their collective reprogramming of transcriptome leads to the enhanced mitochondrial mass, increased number of contact sites between adjacent mitochondrial membranes [193], a higher fusion of the network [194], but also to mitophagy, which is cleaning previously oxidized fragments of mitochondrial network [195]. Regular exercise leads to an elevated density of mitochondrial cristae in trained athletes [196].

Training does not only increase the mitochondrial mass, but also improves function of OXPHOS machinery. In contrast, the acute production of mitochondrial ROS during exercise decreases upon chronic exercise training [197]. It remains to be analyzed whether such a lower amplitude redox signaling takes places at the less background (basal) superoxide formation hence compared to higher amplitude of redox signal, which may probably be superimposed on the higher superoxide background in untrained individuals. Moreover, mitochondrial redox signaling also evokes repair of injured muscles [31]. Mitochondrial redox signaling activates guanosine triphosphatase (GTPase) RhoA, which subsequently triggers accumulation of F-actin, beneficial for the repair of local injury. A self-standing field beyond the scope of this article represents studies of aged skeletal muscle and relations to a sedentary life-style [32]. We should at least note that overexpression of PGC1α in aging muscle led to a reversal of the observed age-related changes [33].

### 6.3. Mitochondrial Network in Skeletal Muscle

We should note that even in skeletal muscle myofibrils, mitochondria form a nearly connected network, such as in other cell types [198] governed by a sophisticated machinery of pro-fusion and pro-fission proteins. Among pro-fusion proteins, OPA1 and MFN2 are upregulated via PGC1α [34]. One can recognize a more fragmented subsarcolemmal part of mitochondrial network and predominantly continuous inter-myofibrillar part. Nevertheless, both parts are interconnected [199]. Whereas the former part provides ATP for active membrane transport and gene transcription, the latter supplies ATP for contractile filaments and is in close contacts with Ca^2+^release units of sarcoplasmic reticulum [200]. Moreover, Ca^2+^-dependency of certain network segments along microtubules is enabled by proteins Kif5B and dynein [201].

### 6.4. Mechanisms of Superoxide Elevation for Mitochondrial Redox Signaling in Skeletal Muscle

One of the mechanisms acutely elevating mitochondrial superoxide formation upon exercise is based on Ca^2+^uptake into the mitochondrial matrix via MCU [31]. For the resting skeletal muscle, one may consider relevant simulations such as studies using specific site inhibitors S1QELs, S3QEL, which reported 12% contribution of Complex I site I_Q_ and 30% contribution of Complex III site III_Qo_ to mitochondrial H_2_O_2_ release in C2C12 myoblasts [202]. After their differentiation, the fraction contribution of Complex I site I_Q_ was 24% but the overall mitochondrial H_2_O_2_ release increased 5-fold. Another mechanism is related to fatty acid β-oxidation elevating superoxide formation in ETFQOR, which may be regarded as initiation of the mitochondrial redox signaling.

## 7. Mitochondrial Signaling in Pancreatic β-Cells

### 7.1. Mitochondrial Signaling during Fatty Acid Stimulated Insulin Secretion in Pancreatic β-Cells

Fatty acids were recognized as so-called insulin secretagogues [29,203,204]. The latter term is used for species stimulating secretion of insulin in pancreatic β-cells. Originally, it has been considered that rather low doses of fatty acids merely amplify the glucose-stimulated insulin secretion (GSIS) [205,206,207]. However, fatty acid-stimulated insulin secretion (FASIS) takes place at low glucose levels that alone do not stimulate insulin release (net GSIS) [203,208]. One can postulate the independence of FASIS as related to high glucose doses (those stimulating insulin release) due to the existence of fatty acid stimulation of metabotropic GPR40 receptor [209,210,211]. The GPR40 pathway acts downstream via the Gq protein or Gs protein, plus arresting routes [212,213,214], and it is predominantly independent off closing of the ATP-sensitive K^+^ channel (K_ATP_) [215]. In contrast, closure of K_ATP_ is the key mechanism involved in GSIS, despite a high amplification ability of incretins, such as GLP1 and GIP, augmenting GSIS. Note also that physiologically fatty acids are delivered to pancreatic β-cells in lipid form within chylomicrons and 2-monoacyl glycerol acts also via a metabotropic receptor GPR119, being a part of so-called glycerol/fatty acid cycle. The latter stimulates insulin secretion partly by OXPSHOS and partly via the exocytosis-promoting protein Munc13-1 activated by the released 2-monoacyl glycerol [204].

Besides the GPR40 pathway of FASIS, a portion of fatty acids enters β-oxidation in mitochondria and is metabolized to increase OXPHOS and hence ATP levels [216]. Nevertheless, as predicted and experimentally verified, β-oxidation of fatty acids generates superoxide which after conversion to H_2_O_2_ serves as a redox signal [45]. This redox signal has been associated with stimulation of insulin secretion in frame of FASIS.

### 7.2. Intramitochondrial Signaling to iPLA2γ Amplifies GPR40 Response

Nevertheless, besides a hypothetical direct redox-sensitive insulin secretion, amplification of GPR40 pathway is provided by the intramitochondrial redox signaling. This is mediated by H_2_O_2_ resulting from ETFQOR-formed superoxide. H_2_O_2_ directly activates mitochondrial Ca^2+^ independent phospholipase A2 isoform γ (iPLA2γ). Subsequently, activated iPLA2γ cleaves fatty acids from phospholipids constituting mitochondrial membranes. The cleaved fatty acids migrate to the plasma membrane, where they additionally stimulate the GPR40 pathway of FASIS [45]. Note that such amplification accounts for ~60% of FASIS-generated insulin.

### 7.3. Mitochondrial Signaling Insulin Secretion Stimulated by Branched Chain Keto-Acids in Pancreatic β-Cells

Also, branched chain (BC) keto-acids are well known insulin secretagogues. They are metabolites of leucine, isoleucine, and valine imported to the mitochondrion. These BC amino acids are converted to 2-keto-isocaproate, 2-keto-3-methyl valerate, and 2-keto-isovalerate, respectively, by mitochondrial BC-amino acid transferase (BCAT) [217]. Next, BC-ketoacid dehydrogenase (BCKDH) converts them to isovaleryl-CoA, 2-methylbutyryl-CoA, and isobutyryl-CoA, respectively. In the subsequent reaction series resembling β-oxidation, final products are made as acetyl-CoA (acetoacetate), propionyl-CoA (or again acetyl-CoA), and succinyl-CoA, respectively, entering the Krebs cycle and being oxidized by the ETFQOR. Subsequently, the elevated OXPHOS results in the K_ATP_-dependent insulin secretion (Figure 5). Mitochondrial requirement for BC-ketoacids as insulin secreatagogues is also supported by the finding that ablation of mitochondrial transcription factor B2 abolishes BC-ketoacid-stimulated secretion of insulin [218].

However, recently we found that insulin secretion stimulated by BC-keto-acids is completely abolished with mitochondrial matrix-targeted antioxidant SkQ1 in insulinoma INS-1E cells [28]. This means that resulting elevation of ATP due to β-like oxidation of BC-keto-acids is unable to close K_ATP_ and hence that parallel redox signaling from mitochondria is essentially required (Figure 5). Similarly as for fatty acids, such redox signaling stems from the elevated superoxide formation with participation of ETFQOR. This conclusion is supported by the finding that silencing of BCKDH largely prevents mitochondrial superoxide formation after 2-keto-isocaproate addition to INS-1E cells and blocks nearly completely resulting secretion of insulin in parallel.

## 8. Mitochondrial Signaling in Immune Cells

### 8.1. Mitochondrial Role and Signaling in Innate Immune Cells

Innate immune cells such as macrophages, neutrophils, monocytes, and dendritic cells coordinate their immune adaptations via distinct metabolic modes, in which mitochondria serve as an essential metabolic and control hub [219,220,221,222]. Thus, mitochondria of innate immune cells provide a signaling center [223] for their specific germline-encoded immune receptors, termed pattern-recognition receptors. Several classes are distinguished, such as toll-like receptors (TLRs) [224]; NOD-like receptors (NLR, e.g., NLRP3, i.e., NOD-like receptor containing pyrin domain 3), forming large cytosolic protein complexes, inflammasomes [225,226]; C-type lectin receptors (CLR), recognizing microorganisms and tumor cells [227]; and RIG-like receptors as sensors of nucleic acids [228], which also include cyclic guanosine monophosphate-adenosine monophosphate (cGAS) [229].

Interestingly, activation of pattern-recognition receptors combined with distinct metabolic modes is integrated to mitochondrial superoxide formation at specific sites. Thus, proinflammatory M1 macrophages undergo transition from OXPHOS to aerobic glycolysis under stimulation by lipopolysaccharide (LPS) plus interferon IFNγ [75]. Thus, pyruvate is largely utilized for lactate production, while glutaminolysis feeds the Krebs cycle with 2OG. In contrast, upon interleukin IL-4 stimulation resolving M2 macrophages rely on OXPHOS fed by glycolysis and fatty acid β-oxidation, while they induce more glutamine metabolism plus UDP-GlcNAc (factor of protein folding and trafficking). The Krebs cycle of M1 macrophages is broken at succinate dehydrogenase and citrate synthase, which leads to succinate and citrate accumulation [230]. Citrate is also converted to itaconate, a metabolite exerting antimicrobial activity, activating NRF2-mediated expression of antioxidant genes [231] and also inhibiting SHD [232]. Together with the succinate accumulation, this results in the enhanced RET and superoxide formation at Complex I I_Q_ site, providing a redox signaling. This RET source of redox signaling can also activate HIF system. In turn, fatty acid β-oxidation could be a source of redox signaling in M2 macrophages.

We should note that mitochondrial superoxide formation is enhanced upon activation of M1 macrophages for the purpose of providing surplus to ROS provided by inducible NADPH oxidases. This is either induced by peroxisome proliferator-activated receptor-γ co-activator 1β (PGC1β) and estrogen-related receptor-α (ERRα), through an IFNγ/STAT-1 signaling [233]; or upon gram-negative bacteria attack via TLR activation [234].

Redox signaling from mitochondria is required for LPS-induced secretion of IL-6, TNFα, and IL-1β, while the target is MAPK kinase [137] (Figure 6). Here, blockage of ROS-attenuating mitochondrial uncoupling mediated by UCP2 is the trigger, since LPS down-regulates UCP2 through the c-Jun N-terminal kinase and p38 pathways. This further stimulates MAPK and thus amplifies a loop potentiating MAPK pathway activation. Consistent with this, UCP2-deficient macrophages exhibited an enhanced inflammatory state [235,236] characterized also by the increased nitric oxide production and elevated migration ability.

### 8.2. Redox Signaling and Other ROS Effects in the Establishment of Inflammasome

Redox signaling was recognized to participate specifically in the assembly of the NLRP3 inflammasome, for which mitochondrial OMM surface provides an assembly platform lying in the enhanced cardiolipin translocation to OMM [237]. Also, mitochondrial antiviral stimulating protein (MAVS) recruits NLRP3 while coordinating RIG-I- and MDA-5-driven production of type-I interferons and proinflammatory cytokines [238]. Moreover, redox signaling from mitochondrion directly activates the assembly of the NLRP3 inflammasome [39]. In this mechanism, mitochondrial uncoupling protein UCP2 plays a modulating role, since when functioning and providing ROS-attenuating uncoupling of mitochondria, NLRP3 and IL-1β expression decreases [239].

The role of ROS in the initiation of the inflammasome is evident and includes cytosolic, as well as mitochondrial sources [39,240,241,242]. As described above, mitochondrial redox signaling participates in its activation and interaction with the component of the NLRP3 inflammasome recognized in immune cells. Mitochondrial superoxide production is induced either at the recognized sites of respiratory chain complexes; namely, at slow down electron transfer, cytochrome *c* cycling, or upon RET. An example can be given as responses to LPS treatment [39,242,243]. The latter leads to the release of mitochondrial DNA, which is especially supposed to be in an oxidized form capable of interacting with AIM2 plus NLRP3, initiating inflammasome activation [244,245]. Interestingly, the crystal structure analysis of NLRP3 identified conserved disulfide bond sensitive to disturbed redox environment [246]. Moreover, other mitochondrial components and stimuli such as cardiolipin, mitofusin MFN2, ATP, and some other components were shown to converge on NLRP3 besides other cellular compartments [237,247]. However, the formation of the inflammasome is a very complex mechanism depending on specific pathogens and endogenous insults and also requiring many components in a two-stage process of formation (priming, activation). Moreover, besides the most studied NLRP3/ACS/caspase-1 inflammasome in cells of the innate immune system, there are plenty of non-canonical and alternative forms of inflammasomes present in various other cell types.

### 8.3. ROS Signaling and Mitochondrial ROS Related to T Cell Activation

It has been recognized that mitochondrial metabolism and morphology regulate immune cell fate during the immune response. This is especially valid for T cells where the transition from quiescence to activate phase involves metabolic rewiring from catabolic to anabolic state. Thus, naïve CD4+ regulatory and memory T cells exhibit oxidative phosphorylation with a preference for fatty acid oxidation to meet relatively low energy requirements, while activated T cells (CD4+ Th1 and CD8+ effector cells) highly depend on glycolysis and pentose phosphate pathway over oxidative phosphorylation. However, even OXPHOS is highly used, linked to glutamine oxidation in activated T cells. Such an adjusted metabolism is required to cover fast cell growth, clonal expansion and effector function. T cell activation involves in the first-place enhanced proliferation, where low/moderate levels of mitochondrial ROS induce nuclear factor activated T cell (NFAT) activation and mTOR signaling leading to Myc upregulation along with HIF system activation [248,249]. The HIF-mediated transcriptome reprogramming is one of the key factors upregulating enzymes of the glycolytic pathway. Lactate acidification in the tumor environment suppresses activation, thus inhibiting T cell proliferation and cytokine production. Within the first 15 minutes of T cell activation mitochondrial ROS are rapidly increased as apparent from oxidized redox probes [37]. As ROS site within mitochondria was suggested respiratory chain Complex III, which is preferentially required for CD4+ T cell activation and IL2 production [37].

Recently, Chandel and colleagues found that mice lacking Complex III specifically in regulatory T cells displayed a loss of T cell-suppression capacity without altering cell proliferation and survival [221]. Kaminski et al. suggested that mitochondrial Complex I should be induced by PKC as a primary superoxide producer, where immediate conversion to H_2_O_2_ by MnSOD activates expression of T cell activated cytokines IL2 and IL4 [38,250]. As an alternative ROS source, mitochondrial glycerol-3-phosphate dehydrogenase could either form ROS directly or should accumulate reduced ubiquinol, thus stimulating superoxide production within the Complex I [251]. Glutaminolysis and pentose phosphate cycle provide precursors for various biosynthetic pathways and significantly enhance NADPH, thus carefully balancing pro-oxidative status [252]. Enhanced AMPK activity which also inhibits mTOR then significantly increases ROS and reduces aerobic glycolysis [253]. On the other hand, regulatory T cells arising from CD4+ stimulation induce FAO by AMPK activation while enhanced mitochondrial ROS suppress mTOR activation. This represents a similar mechanism to the metabolism of memory T cells. Enhanced expression of lysosomal acid lipase mobilizes stored lipids to fatty acids required for fatty acid oxidation [254]. IL7 is also required for memory T cell activation to mediate triglyceride synthesis and GLUT1 trafficking to the membrane [255,256]. Thus, it is apparent that controlled redox status ensures the integrity of T cell energy metabolism providing proper T cell function. Excessive ROS generation induces senescence and malignant transformation or T cell death involving genes including Bcl2 and FasL.

### 8.4. ROS Signaling and Mitochondrial ROS Related to B Cell Activation

Antigen receptor diversification of naïve B lymphocytes requires alteration of immunoglobulin function by class-switch recombination (CSR) and differentiation into antibody-secreting plasma cells (PCD) or memory B cells [257]. Mitochondrially derived ROS regulates heme synthesis which is required within the CRS process while being suppressed during PCD [258].

## 9. Mitochondria and Kinase Signaling

Mitochondrial proteins are often targets of physiological cell signaling [259], including cardioprotective signaling and prevention of programmed cell death [260]. H_2_O_2_ either directly or in conjunction with peroxiredoxins acts as a predominant intracellular redox-signaling molecule, being able to oxidize catalytic cysteine thiol groups of protein tyrosine phosphatases and numerous protein kinases [2,261]. As discussed above, Janus kinase/JAK plus STAT3 pathway and ASK1 pathway were recognized to be affected by redox signaling (see Section 4.5) (Figure 6). Increasing evidence is also being gathered that indicates the role of the protein kinase C (PKC) family of isoenzymes in transducing H_2_O_2_-induced signaling in a wide variety of physiological and pathophysiological processes [262,263,264,265,266,267].

Moreover, mitochondrial ROS (H_2_O_2_) enhances phosphoinositide3-kinase (PI3K) signaling by deactivating the phosphatase PTEN via cysteine oxidation [268]. Deactivation of PTEN subsequently enables activation of protein kinase B/Akt, serving as a positive regulator of the PI3Kpathway and driving cell proliferation. Mitochondrial ROS (H_2_O_2_), and hence redox signaling, evidently participates in angiotensin II signaling, which is required for downstream signaling cascades (Figure 6). For example, in vascular smooth muscle cells, stimulation with angiotensin II is parallel to the increased H_2_O_2_ release from mitochondria, which further induces MAPK signaling [269].

## 10. Future Perspectives

Further research is required to definitively unravel mechanism(s) on how redox signaling from mitochondria is transferred to the distant target proteins such as those located at the plasma membrane or nucleus. Direct diffusion of particular species such as superoxide and, namely, H_2_O_2_ should be either confirmed or excluded, as well as the mutual transfer of redox state among an array of peroxiredoxins. One should study, whether before a peroxiredoxin “redox kiss” to a target protein, decamers of peroxiredoxin diffuse so as to gain the H_2_O_2_ redox signal further away from the target. If this is not possible, details should be revealed to answer the question of how far H_2_O_2_ (or superoxide) can diffuse to encounter the first peroxiredoxin decamer.

Also, since a vast majority of proteins are an oxidatively modified concept of passive targets and those terminal ones, mediating the subsequent biological response should be verified. Of course, this is expected to be yielded from the genome-wide studies at exemplar situations, together with further detailed identification of new targets. As a result, new redox-regulated biological phenomena should be yielded from all such efforts. A prominent place could be represented by redox regulations of kinase signaling. A threshold between physiological redox signaling vs. redox burst leading to the pathological outcome should be distinguished in detail, as well as both events from permanent oxidative stress.

Concerning mitochondria as a major source of superoxide, the interplay between ion channels or uncoupling proteins and sources of superoxide under distinct conditions should be unraveled, along with mutual relationships between morphology of mitochondrial network or ultramorphology of mitochondrial cristae, both in relation to superoxide formation, as well as to apoptosis initiation, mitophagy, machinery of mitochondrial DNA, etc. Finally, feedback of redox changes initiating in mitochondria should be recognized.

## Figures and Tables

**Figure 1 biomolecules-10-00093-f001:**
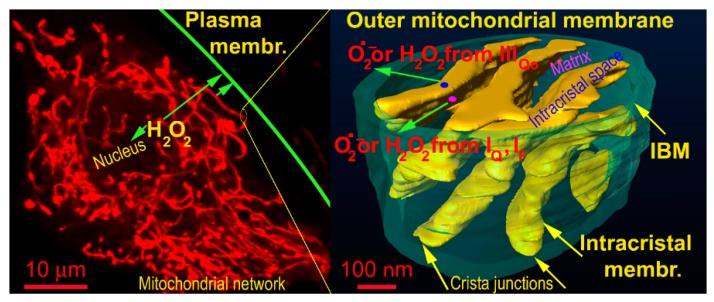
Representation of distances for hypothetic H_2_O_2_ diffusion to plasma membrane — demonstrated on a confocal image of the mitochondrial network (left) and FIB/SEM 3D image of cristae within a segment of the mitochondrial network (oriented vertically). The arrows show diffusion distances from the most proximal and most distant tubule of the mitochondrial network to the plasma membrane (left) and diffusion from the matrix (translucent space inside the IBM cylinder) or from then intracristal space (yellow topology, which also comprises ATP synthase oligomers and respiratory chain supercomplexes).

**Figure 2 biomolecules-10-00093-f002:**
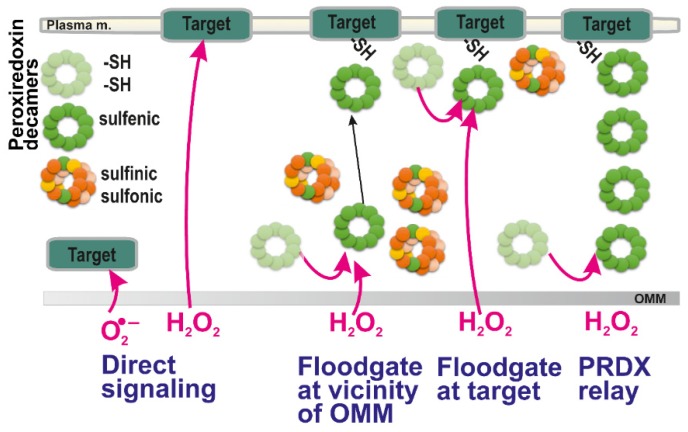
Possible ways of redox signal spreading — from left to right: (*i*) direct superoxide diffusion; (*ii*) direct H_2_O_2_ diffusion; (*iii*) peroxiredoxin-mediated redox signal transfer including diffusion of peroxiredoxin decamers; (*iv*) combination of (*ii*) and (*iii*), i.e., H_2_O_2_ diffusion followed by the peroxiredoxin relay (peroxiredoxin “redox kiss”); and (*v*) hypothetical redox relay via an array of peroxiredoxins. Note, that according to a flood-gate model, H_2_O_2_ oxidizes PRDX to higher states than a sulfenic state (*green*; basic reduced state *light green*), such as sulfinic (*yellow*) and irreversible sulfonic state (*orange*). This allows only distant decamers in a sulfenic state either to migrate to the target (*iii*) or to oxidize target at its vicinity (*iv*). The hypothetical mechanism (*v*) should still be verified experimentally.

**Figure 3 biomolecules-10-00093-f003:**
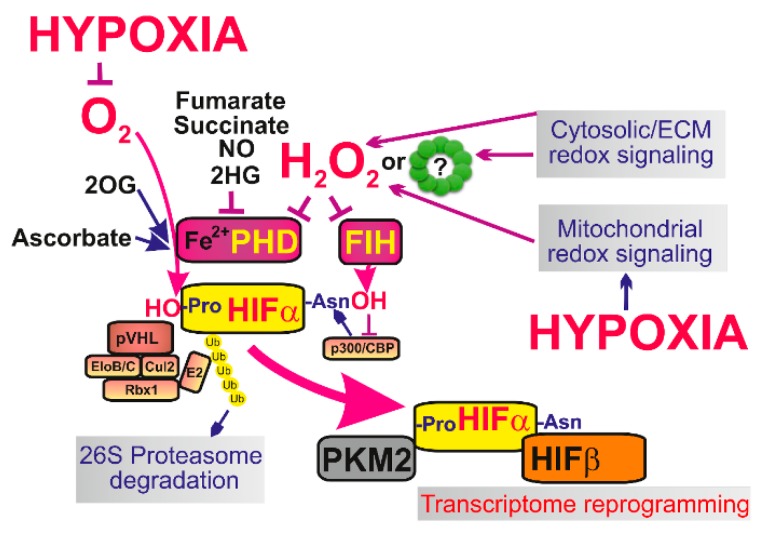
The major mechanisms of initiation of HIF-mediated transcriptome reprogramming—At normoxia isofoms of hypoxia-inducible factor α (HIF-α), such as HIF-1α are constantly degraded which is ensured by ubiquitin ligase Von Hippel-Lindau tumor suppressor protein (pVHL) and the components of the proteasome complex e.g., elongin B/C (EloB/C), cullin 2 (Cul2), ring-H2 finger protein (Rbx1) or ubiquitin ligase E2. During hypoxic adaptation the lack of oxygen and mitochondrial redox signaling (or in some situations also elevation of cytosolic ROS) lead to stabilization of HIF-α and it’s binding to HIF-β. As a result stabilized HIF with the help of transcription factor p300/CBP causes transcriptome reprogramming (up to 400 genes are affected) and a typical metabolic switch to Warburg phenotype, when OXPHOS is suppressed while glycolysis and lactate production is upregulated. The link between low oxygen levels and mitochondrial redox signaling is provided and in fact integrated by the resulting inactivation of prolyl hydroxylases (PHDs) and factor inhibiting HIF (FIH), which no longer are able to hydroxylate HIFα and cause it’s degradation. Pyruvate kinase isozyme M2 (PKM2) migrates to nucleus to specifically enhance transcription.

**Figure 4 biomolecules-10-00093-f004:**
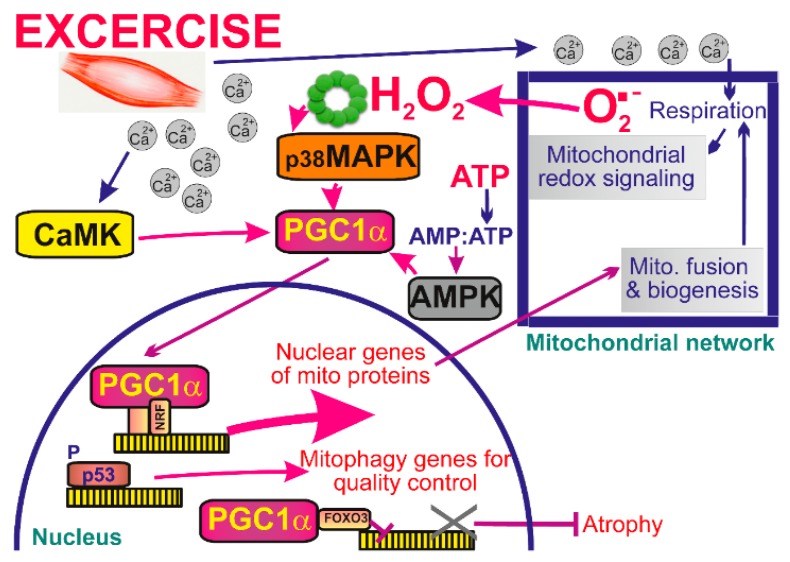
Mitochondrial signaling in skeletal muscle during exercise. During exercise cytoplasmic Ca^2+^ concentration rises leading to activation of calcium-calmodulin-dependent kinase (CaMK) and increase in mitochondrial respiration, and subsequent increase in ATP and ROS production. While Ca^2+^ acts via CaMK, the AMP/ATP ratio stimulates the AMP-activated protein kinase (AMPK) pathway and mitochondrial ROS target the p38 mitogen-activated protein kinase pathway (MAPK). All signals are then integrated in the increase of transcriptional activity of the master regulator of mitochondrial biogenesis, peroxisome proliferator-activated receptor-gamma coactivator-1α (PGC1α).

**Figure 5 biomolecules-10-00093-f005:**
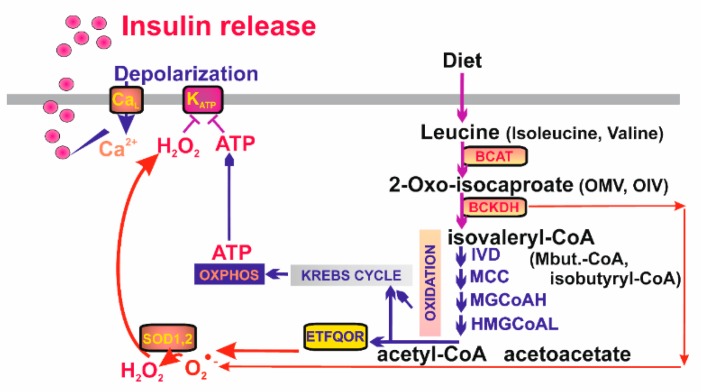
Mitochondrial redox signaling as essential part of branched-chain keto-acid- stimulated secretion of insulin. Branched-chain keto-acids are metabolized by a series of catabolic reactions leading to production of substrates fueling Kreb‘s cycle and the mitochondrial respiratory chain. ATP and H_2_O_2_ then cause closure of the ATP-dependent K^+^ channels (K_ATP_) on the plasmatic membrane, membrane depolarization and activation of the voltage-gated Ca^2+^ channels (Ca_L_) setting off the canonical pathway of insulin release. BCAT—branched-chain α-ketoacid amino transferase; BCKDH—branched-chain α-ketoacid dehydrogenase; IVD—isovaleryl-CoA dehydrogenase; MCC—methylcrotonyl-CoA carboxylase; MGCoAH—methyl-glutoconyl-CoA hydratase; HMGCoAL—3-hydroxy-3-methylglutaryl-CoA lyase.

**Figure 6 biomolecules-10-00093-f006:**
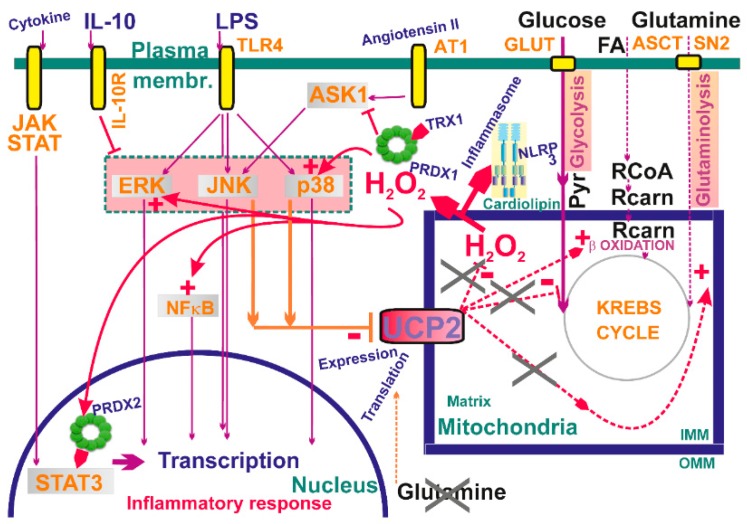
Examples of mitochondrial signaling upon activation of different kinases—schematic overview examples of established participation of mitochondria-generated H_2_O_2_ in signaling cascades. Also, a role of inhibition of UCP2 is outlined, resulting in elevation of mitochondrial superoxide generation and hence elevated H_2_O_2_ release. For a detailed explanation see Section 8 and Section 9.

**Table 1 biomolecules-10-00093-t001:** Exemplar physiological situations with mitochondrial redox signaling and possible pathological consequences of its exaggeration and/or impairment.

Source/Event	Physiological Target/Function	Ref.	Source/Event	Pathology	Ref.
MitoROS = redox signaling/hypoxia	PHD/HIF-mediated transcriptome reprogramming	[19][20][21]	MitoROS - PHD-HIF - Warburg phenotypeOncogenes2hydroxyglutarate- altered epigenetics	Cancer	[2,8,22][23,24][25]
RET, K_ATP_ opening => mito ROS	NOX2 in endothelial cells	[26]	RET, frequent K_ATP_ opening => mito ROS	Endothelial cell OX.STRESS Hypertension	[26]
MitoROS	NOX4 in pulmonary endothelial and recruited immune cells, fibroblasts	[6][27]	MitoROS - PHD-HIF - Warburg phenotype	Pulmonary arthery remodellingPulmonary hypertension	[6][27]
MitoROS = redox signaling	Plasma membrane K_ATP_ closure=> insulin release	[28]	Impaired Mito redox signalingOX.STRESS in pancreatic β-cells	Type 2 diabetes	[29]
MitoROS= redox signaling, skeletal muscle at excercise	PGC1α, skeletal muscle rejuvenation	[30][31]	Impaired Mito redox signalingOX.STRESS, sedentary life-style	skeletal muscle senescence, weakness, athrophy	[32][33,34]
			Succinate accumulation => RET	Hypoxia/reperfusion Indry (heart)	[35]
Succinate accumulation => RET => intramitochondrial redox signaling	UCP1 in brown adipose tissue/thermogenesis	[14,36]			
MitoROS = redox signaling in T cells	NFAT, NFκB/Proximal T cell receptor signaling	[37][38]			
MitoROS = redox signaling, immune cells	NLRP3 inflammasome/IL-1β secretion	[39]

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
