# Peer review of "Redox Signaling from Mitochondria: Signal Propagation and Its Targets"

_biomolecules, 2020, doi:10.3390/biom10010093_

Round 1

Reviewer 1 Report

Cellular redox homeostasis, reactive oxygen species from mitochondria study is very well studied subjects and widely used terminology in health and disease.

Broad Comments:

Overall, this paper includes comprehensive description. Minor suggestion could be considered in this review: The title needs to be more focus on the overall elaboration throughout the paper.

This is a manuscript of review. So the overall organization and visualization are key points for readers to make it easy to understand and follow. It would be helpful if authors can make tables to connect redox to the related pathways and phenotype of human physiopathology.

This article has comprehensively reviewed numerous molecular mechanisms regarding redox. I recommend that the authors remove the less precise section 8 and 9 descriptions.

Author Response

We thank the reviewer for improving our review manuscript.

Reviewer #1: Comment 1. Overall, this paper includes comprehensive description. Minor suggestion could be considered in this review: The title needs to be more focus on the overall elaboration throughout the paper.

Our response: The title was changed to „Redox signaling from mitochondria: signal propagation and its targets

Reviewer #1: Comment 2. This is a manuscript of review. So the overall organization and visualization are key points for readers to make it easy to understand and follow. It would be helpful if authors can make tables to connect redox to the related pathways and phenotype of human physiopathology.

Our response: Table 1 was amended as requested by the reviewer. 

Reviewer #1: Comment 3. This article has comprehensively reviewed numerous molecular mechanisms regarding redox. I recommend that the authors remove the less precise section 8 and 9 descriptions.

Our response: The section 8 and section 9 were deleted.

Reviewer 2 Report

This paper overview the Redox signaling from mitochondria, the mitochondrial sources of reactive oxygen species, their targets inside of mitochondria, their reactions in the cytosol and their effects on the nuclei function. This work provides balanced and up-to-date review of the subject which will be of potential interest for a wide audience.

Minor comments:

1) Please make corrections on page 7, Section 2.1 Hypothetical redox signaling by superoxide diffusion?

Authors state that "half-life of dismutation at 1uM O2•− is 10 s at pH 7."

This is not quite correct. Please cite the paper of Bielski B.H.J. and Allen A.O. Mechanism of the Disproportionation of Superoxide Radicals. The Journal of Physical Chemistry, Vol. 81, No. 11, 1977.

They showed that the dismutation rate constant at pH 7 is 500.000, therefore, the half-life of dismutation at 1uM superoxide is 1/{10(-6)*5(10(5)}=2 second  

Please add the reference and update the text.

2) Authors are advised to add a small section or paragraph regarding the potential role of Redox signaling from mitochondria to activate the NADPH oxidases such as Nox1 and Nox2 via c-Src redox dependent mechanism (Antioxid Redox Signal. 2014;20(2):281-94; Am J Physiol Heart Circ Physiol. 2013;305(8):H1131-40).

The potential cross-talk between mitochondrial ROS and NADPH oxidase have been previously described (Free Radic Biol Med. 2011;51(7):1289-301).

Author Response

We thank the reviewer for improving our manuscript.

Reviewer #2: Comment 1. Please make corrections on page 7, Section 2.1 Hypothetical redox signaling by superoxide diffusion?Authors state that "half-life of dismutation at 1uM O2•− is 10 s at pH 7." This is not quite correct. Please cite the paper of Bielski B.H.J. and Allen A.O. Mechanism of the Disproportionation of Superoxide Radicals. The Journal of Physical Chemistry, Vol. 81, No. 11, 1977. They showed that the dismutation rate constant at pH 7 is 500.000, therefore, the half-life of dismutation at 1uM superoxide is 1/{10(-6)*5(10(5)}=2 second. Please add the reference and update the text.

Our response: We mentioned 2 s as a half-life of dismutation at 1uM superoxide and amended the suggested reference.

Reviewer #2: Comment 2. Authors are advised to add a small section or paragraph regarding the potential role of Redox signaling from mitochondria to activate the NADPH oxidases such as Nox1 and Nox2 via c-Src redox dependent mechanism (Antioxid Redox Signal. 2014;20(2):281-94; Am J Physiol Heart Circ Physiol. 2013;305(8):H1131-40). The potential cross-talk between mitochondrial ROS and NADPH oxidase have been previously described (Free Radic Biol Med. 2011;51(7):1289-301).

Our response: We added a new section 3.8 entitled : “Amplification of cytosolic ROS production (signaling) by stimulation of NADPH oxidases by mitochondrial ROS“ and inserted more references than suggested by the reviewer.